# A Rare Case of Recurrent Generalized Peritonitis Caused by Spontaneous Urinary Bladder Rupture after Radiotherapy: A Case Report and Literature Review

**DOI:** 10.3390/medicines8110067

**Published:** 2021-11-05

**Authors:** Yusuke Watanabe, Shun Yamazaki, Hanako Yokoyama, Shunta Yakubo, Akihiko Osaki, Kenichi Takaku, Munehiro Sato, Nobuo Waguri, Shuji Terai

**Affiliations:** 1Department of Gastroenterology and Hepatology, Niigata City General Hospital, Niigata 950-1141, Japan; shuny1992@med.niigata-u.ac.jp (S.Y.); hanakoy87@outlook.jp (H.Y.); sh.yakubo@gmail.com (S.Y.); a.osaki@hosp.niigata.niigata.jp (A.O.); wcyty355@gmail.com (K.T.); munesato0428@yahoo.co.jp (M.S.); waguri@hosp.niigata.niigata.jp (N.W.); 2Division of Preemptive Medicine for Digestive Disease and Healthy Active Life, School of Medicine, Niigata University, Niigata 951-8510, Japan; 3Division of Gastroenterology and Hepatology, Graduate School of Medical and Dental Sciences, Niigata University, Niigata 951-8520, Japan; terais@med.niigata-u.ac.jp

**Keywords:** generalized peritonitis, urinary ascites, spontaneous urinary bladder rupture, radiotherapy

## Abstract

Since generalized peritonitis is a fatal disease, accurate diagnosis and treatment are important. In this paper, we report a case of recurrent generalized peritonitis associated with spontaneous urinary bladder rupture (SBR). A 65 year old woman, who underwent radiotherapy 21 years prior, was diagnosed with generalized peritonitis. Although the cause of the generalized peritonitis could not be identified, the patient recovered with conservative treatment in short period. However, recurrent episodes of generalized peritonitis occurred four times. We diagnosed the patient with urinary ascites due to SBR, based on a history of radiotherapy and dysuria. No recurrence of generalized peritonitis had occurred after accurate diagnosis and treatment with long-term bladder catheter placement. Since SBR often occurs as a late complication after radiotherapy, it is difficult to diagnose SBR, which leads to delayed treatment. This case and literature review of similar cases suggest that the information of the following might be helpful in the diagnosis of SBR: (i) history of recurrent generalized peritonitis, (ii) pseudo-renal failure, (iii) history of radiotherapy, (iv) dysuria, and (v) increase or decrease of ascites in a short period. It is important to list SBR in the differential diagnosis by knowing the disease and understanding its clinical features. This case and literature review will serve as a reference for future practices.

## 1. Introduction

Generalized peritonitis is a condition in which inflammation occurs and spreads into the peritoneum [1]. The causes of generalized peritonitis can be classified as the following: primary, such as spontaneous bacterial peritonitis in cirrhotic patients; secondary, such as gastrointestinal perforation, intraperitoneal rupture of malignant tumors, and Fitz-Hugh–Curtis syndrome; tertiary, such as in the case of an intra-abdominal abscess [1,2,3,4,5]. Depending on the cause, diagnosis is based on a comprehensive evaluation of vital signs, physical examination, laboratory findings, or imaging [1]. In the physical examination, generalized peritonitis patients have symptoms of severe abdominal pain, peritoneal irritation, and rebound tenderness [1]. In the laboratory findings, generalized peritonitis patients have a high degree of inflammation, and the imaging of generalized peritonitis patients are important to identify free-air and a perforation area, especially in secondary peritonitis [6].

The treatments for generalized peritonitis also depend on the causes of peritonitis. In the primary and tertiary peritonitis, antibiotics are mainly used, and drainage therapies (such as ascites drainage or abscess drainage) are added depending on the situation [4,7]. In the secondary peritonitis, surgery is performed, including suturing and omentoplasty to the perforation area [2]; however, in cases of small perforations, for example, conservative treatment with antibiotics would be performed to avoid invasive surgery [8]. Since the treatment of generalized peritonitis varies due to the causes, proper diagnosis is important. In severe cases of generalized peritonitis, patients have sepsis, septic shock, disseminated intravascular coagulation, and multiple organ failure, which require intensive care treatment; however, the prognosis might be poor, with high mortality rates [9,10,11,12].

Causes of generalized peritonitis include secondary peritonitis associated with spontaneous bladder rupture (SBR) [13,14,15,16]. SBR is a rare disease caused by trauma, radiotherapy, chronic bladder infection, bladder tumor, or bladder diverticulum [13,14,15,16]. Surgery, or conservative treatment such as bladder catheter placement, is performed as a treatment for SBR [17]. Physicians do not have the opportunity to treat SBR frequently; therefore, it is difficult to diagnose SBR [13,14,15,16].

Herein, we report a rare case of generalized peritonitis associated with SBR, which was difficult to identify as the cause of recurrent episodes of abdominal pain. In addition, we describe the clinical features of this case and similar cases.

## 2. Case Report

A 65 year old woman, who had undergone a radical hysterectomy and postoperative radiotherapy (total dose, 50 Gy) for uterine endometrial cancer 21 years prior, was admitted to our hospital with acute abdominal pain. The patient had symptoms of peritoneal irritation with rebound tenderness and was suspected to have generalized peritonitis with inflammatory findings (white blood cell (WBC), 11,000/µL; neutrophils, 85.2%; C-reactive protein (CRP), 3.50 mg/dL; platelet, 30.7 × 10⁴/µL; procalcitonin, 17.35 ng/mL) and renal failure (blood urea nitrogen (BUN), 52.3 mg/dL; creatinine (Cre), 3.73 mg/dL). Plain computed tomography (CT) showed only massive ascites, but showed no findings of gastrointestinal perforation, such as free air. Ascites collected by puncture were neither purulent nor bloody. Conservative treatment was performed because of stable vital signs and the patient’s will. Considering bacterial peritonitis, the antibiotics tazobactam/piperacillin (7.75 g/day) were administered. On day 3 after admission, the abdominal symptoms and inflammatory findings improved (WBC, 4800/µL; Neutrophils, 59.6%; CRP, 0.71 mg/dL). On day 4 after admission, renal failure improved (BUN, 9.1 mg/dL; Cre, 0.53 mg/dL), and contrast-enhanced CT was performed, showing no remarkable findings for the cause of generalized peritonitis (Figure 1). In addition, no abnormalities were found on esophagogastroduodenoscopy or total colonoscopy. Recurrence of uterine endometrial cancer was suspected; however, tumor markers did not elevate (carcinoembryonic antigen (CEA), 0.9 ng/dL; carbohydrate antigen 19-9 (CA19-9), 6.6 u/mL), and a CT showed no disseminated nodules. The tumor markers and adenosine deaminase (ADA) in the ascites also did not elevate (CEA, 0.5 ng/dL; CA19-9, 3.2 U-mL; ADA, 2.1 u/L). The cytology of ascites was class II and was dominated by WBC (especially neutrophils), reflecting generalized peritonitis. Although the cause of the generalized peritonitis could not be identified, conservative treatment, including intravenous fluid treatment without eating, in addition to antibiotics, led to the disappearance of ascites and the patient’s discharge on day 15.

Five months after the first admission (second-time admission), the patient was readmitted for the same symptoms. The cause was still unclear; however, the patient recovered with conservative treatment. Eleven months after the first admission (third-time admission), the patient was readmitted for the same episode of acute abdominal pain.

To investigate the cause of generalized peritonitis, a trial laparotomy was suggested; however, the patient refused the procedure. Fourteen months after the first admission, the patient was readmitted for the fourth time with generalized peritonitis. A plain CT, on the day of the fourth admission, showed massive ascites and partial wall thinning in the dome of the bladder (Figure 2). A urine culture revealed a urinary tract infection (Klebsiella oxytoca detected) along with urinalysis (WBC in urine, 2+). Since the patient complained of dysuria, a cystoscopy was performed just after the inflammatory findings improved. The cystoscopy showed no tumor or fistula but showed granular edematous mucosa in the same part of wall thinning on CT at the fourth admission (Figure 3). Based on these findings, SBR was diagnosed as the cause of recurrent generalized peritonitis. We suggested surgery of suturing and omentoplasty to the wall thinning in the dome of the bladder; however, the patient declined it because of the invasiveness. Instead of the surgery, the patient received a long-term bladder catheter placement, leading to no recurrence of generalized peritonitis for 2 years.

## 3. Discussion

SBR is a rare cause of secondary peritonitis, which needs to be differentiated from primary peritonitis (such as spontaneous bacterial peritonitis) and tertiary peritonitis (such as intra-abdominal abscess) [1,2,3,4,5,11,18]. Accurate diagnosis and treatment for SBR are important; however, if left undiagnosed and untreated, SBR is a recurring condition, leading to a poor prognosis [19]. Since SBR often occurs as a late complication after radiotherapy, it is difficult to diagnose SBR, leading to delayed treatment [20]. Since approximately 2.1% of patients have bladder injury after radiotherapy for gynecologic malignancies [21], it is important to pay attention to patients with a history of radiotherapy (especially for gynecologic conditions). The biological mechanisms for SBR occurrence after radiotherapy is radiation-induced wall weakness. Radiation causes pathological changes in the bladder wall, such as epithelial cell damage, submucosal fibrosis, and interstitial changes [22]. Along with these pathological changes, the dome of the bladder, especially with its thin muscular layer and lack of supportive tissue, often ruptures during intravesical pressure increasing [23]. These pathological and anatomical features of radiation lead to SBR occurrence. In this case, SBR was suspected to be the cause of recurrent generalized peritonitis. The injury in the dome of the bladder, in this case, occurred 21 years after radiotherapy, which is consistent with previous reports. In addition, a previous study reported that recurrent urinary tract infection might occur asymptomatically because of loss of bladder sensation after hysterectomy [24]. Based on this reference, in the present case, since the ascites cells were mainly neutrophils and the urine culture showed Klebsiella oxytoca, bacterial transfer with urine from chronic asymptomatic cystitis might be the cause of the recurrent generalized peritonitis. Cultures in ascites showed no detectable bacteria, which might be due to the small amount of bacterial transferred and the use of antibiotics.

The diagnostic methods for SBR are cystography or cystoscopy, and confirming the presence of a fistula is most useful [25,26,27,28,29,30,31]. CT can demonstrate the findings of SBR in the presence of contrast medium in the abdominal cavity after intravenous contrast medium injection [32] and abnormalities of the bladder wall on CT [33]. However, sometimes a bladder spasm or intestinal covering closes the fistula of the bladder wall, which makes it difficult to identify the fistula [34]. In some cases, ascites increase or decrease in a short period, which might be helpful in diagnosis [35,36]. In this case, cystoscopy did not confirm a fistula in the bladder; however, SBR was suspected because of granular edematous mucosa on cystoscopy and thinning of the bladder wall on CT. Furthermore, the increase or decrease of ascites in a short period was consistent with SBR. During cystoscopy, the patient complained of severe abdominal pain. Therefore, considering the invasiveness, additional cystography could not be performed. From this, our diagnosis for SBR was based on a comprehensive evaluation of cystoscopy findings, ascites findings, and CT image findings.

SBR is treated conservatively or with surgery, as described above. In cases where the fistula of the bladder is not confirmed by cystography, cystoscopy, or CT, conservative treatment might be an option for the treatment of SBR [34]. In this case, since a fistula in the bladder was not confirmed, conservative treatment was performed. We suspected that the fistula was probably closed when examined by cystoscopy, and the fistula opened again during the intravesical pressure increasing. Initially, considering the overall survival of the patient, surgery might be preferable, however, the patient declined the surgery because of the invasiveness. Although long-term bladder catheter placement had the risk of urinary tract infection or recurrent bladder rupture [37], the patient underwent insertion of a bladder catheter, leading to no recurrence.

SBR causes generalized peritonitis, leading to sepsis. Intensive care treatment is required for sepsis, including treatment of the cause, antibiotics, corticosteroids, and catecholamines [38,39]. In this case, although we suspected the occurrence of sepsis based on high procalcitonin level, and renal failure, the patient did not have a septic shock and responded well to intravenous fluid treatment. Although the sepsis-related organ failure assessment score of this patient is three points due to renal failure [40], considering the general condition, we performed antibiotic usage only.

In this case, serum BUN and Cre levels were elevated during the all episodes of generalized peritonitis. The serum level of BUN and Cre in each admission was 52.3 mg/dL, 3.73 mg/dL (first admission), 41.0 mg/dL, 2.07 mg/dL (second admission), 62.8 mg/dL, 3.12 mg/dL (third admission), and 44.7 mg/dL, 2.81 mg/dL (fourth admission), respectively. There is a possibility of renal failure due to sepsis, prerenal failure due to dehydration, and postrenal failure due to urinary retention; however, based on facts such as the rapid improvement of renal failure, the lack of dehydration in laboratory findings, and no hydronephrosis on CT, these possibilities are unlikely in this case [41,42]. Renal failure might be due to reverse dialysis, in which urinary Cre and BUN in the peritoneal cavity are reabsorbed by the peritoneum [43,44]. Pseudo-renal failure associated with reverse dialysis is one of the findings of SBR [43,44], and this finding might help to diagnose SBR.

A literature search using the terms “spontaneous urinary bladder rupture”, “radiotherapy”, and “peritonitis” in PubMed showed that only 25 cases (including this case) of generalized peritonitis, caused by SBR after radiotherapy, have been reported, and the available information is summarized in Table 1 [25,26,27,28,29,30,31,32,35,36,45,46,47,48,49,50,51,52,53,54,55,56]. Previously reported patients included 5 men and 20 women, ranging from 27 to 82 years (median: 64 years) (3 cases unknown). The time after radiotherapy ranged from 1 to 22 (median: 10) years—this result was consistent with SBR occurrence late after radiotherapy. Fistulas were confirmed in 10 cases and not confirmed in 11 cases (4 cases unknown), suggesting that it was difficult to diagnose SBR based on fistula confirmation. SBR might be comprehensively diagnosed based on other clinical symptoms, such as reverse dialysis. Surgery in 10 cases and conservative treatment in 13 cases (2 cases unknown) were performed. Both treatments led to survival outcomes in all cases, which showed that accurate diagnosis and treatment for SBR might improve the prognosis.

In summary, it is important to know the disease and understand its clinical features of SBR as a differential diagnosis. If a fistula cannot be identified, the following factors might be helpful in the diagnosis of SBR: (i) the patient’s history of recurrent generalized peritonitis; (ii) transient elevation of Cre and BUN during the episode of generalized peritonitis (pseudo-renal failure); (iii) the patient’s history of radiotherapy (especially for gynecologic conditions); (iv) dysuria; (v) increase or decrease of ascites in a short period. SBR is a rare condition; however, with medical advances, the number of SBR cases is expected to increase as more patients survive long term after radiotherapy. This case report and literature review will serve as a reference for future practices.

## 4. Conclusions

We report a rare case of recurrent generalized peritonitis caused by SBR. SBR should be included as a cause of generalized peritonitis in patients with a history of radiotherapy.

## Figures and Tables

**Figure 1 medicines-08-00067-f001:**
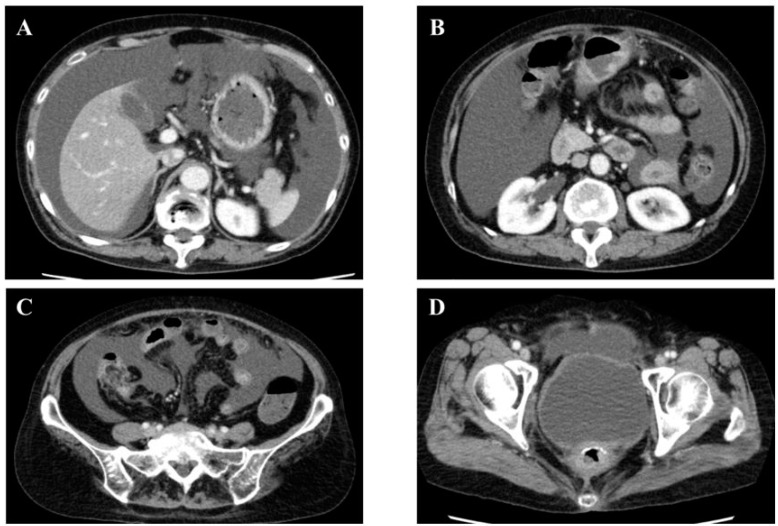
Abdominal contrast-enhanced computed tomography on day 4 of 1st admission. (**A**–**D**): Abdominal contrast-enhanced computed tomography showed only massive ascites. There were no gastrointestinal perforations, malignant tumors, or urinary abnormalities.

**Figure 2 medicines-08-00067-f002:**
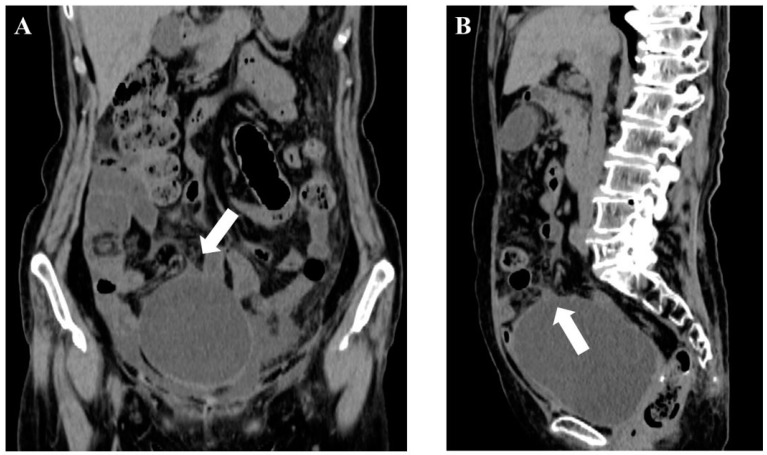
Abdominal plain computed tomography on the day of 4th admission. Abdominal plain computed tomography showed massive ascites and partial wall thinning (white arrow) in the dome of the bladder. (**A**) coronal image; (**B**) sagittal image).

**Figure 3 medicines-08-00067-f003:**
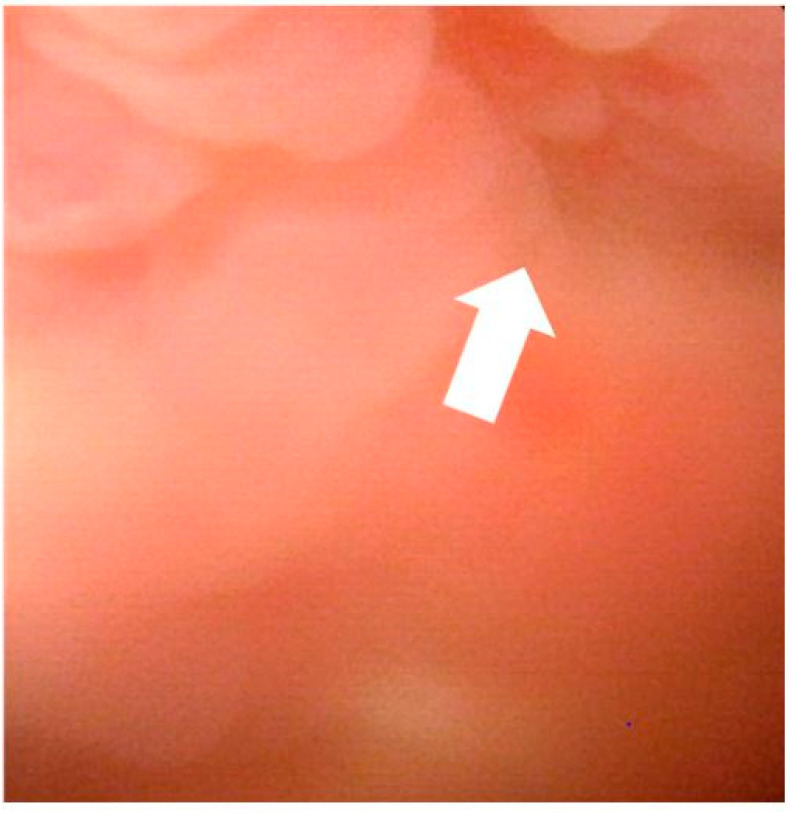
Cystoscopy on the day of 4th admission. Cystoscopy showed no tumor or fistula, but showed granular edematous mucosa in the same part of wall thinning on computed tomography at 4th admission (white arrow).

**Table 1 medicines-08-00067-t001:** Summary of 25 cases of spontaneous bladder rupture after radiotherapy. CT—computed tomography; n.a.—not applicable; No.—number; SBR—spontaneous bladder rupture.

No.	Age	Gender	Treatment for SBR	Prognosis	Years after Radiotherapy	Presence or Absence of Fistula	Diagnosis	Reference No.
1	44	female	conservative therapy	alive	10 years	presence	cystoscopy	[25]
2	27	female	conservative therapy	alive	within 1 year	presence	cystography	[26]
3	53	female	surgery	alive	4 years	presence	cystography	[27]
4	65	female	surgery	alive	7 years	presence	cystography	[27]
5	54	female	conservative therapy	alive	3 years	presence	cystoscopy	[27]
6	67	female	surgery	alive	21 years	presence	cystography	[28]
7	64	female	conservative therapy	alive	7 years	presence	cystoscopy	[29]
8	55	female	surgery	alive	13 years	presence	cystography	[30]
9	74	female	surgery	alive	11 years	presence	cystography	[31]
10	61	female	surgery	alive	within 1 year	presence	CT	[32]
11	68	female	conservative therapy	alive	13 years	absence	cystoscopy	[35]
12	67	male	surgery	alive	within 1 year	absence	cystography	[36]
13	68	female	conservative therapy	alive	22 years	absence	cystography	[45]
14	65	male	conservative therapy	alive	2 years	absence	cystoscopy	[46]
15	n,a.	female	n.a.	alive	3 years	absence	cystography	[47]
16	62	female	conservative therapy	alive	17 years	absence	cystography	[48]
17	54	female	conservative therapy	alive	16 years	absence	cystoscopy	[49]
18	82	male	surgery	alive	within 1 year	absence	CT	[50]
19	44	female	surgery	alive	5 years	absence	CT	[51]
20	79	female	surgery	alive	11 years	absence	CT	[52]
21	n.a.	male	conservative therapy	alive	17 years	n.a.	n.a.	[53]
22	n.a.	female	conservative therapy	alive	15 years	n.a.	n.a.	[54]
23	n.a.	male	conservative therapy	alive	18 years	n.a.	n.a.	[55]
24	27	female	n.a.	alive	10 years	n.a.	n.a.	[56]
25	65	female	conservative therapy	alive	21 years	absence	CT and cystoscopy	our case

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
