# Peer review of "A Rare Case of Recurrent Generalized Peritonitis Caused by Spontaneous Urinary Bladder Rupture after Radiotherapy: A Case Report and Literature Review"

_medicines, 2021, doi:10.3390/medicines8110067_

Round 1

Reviewer 1 Report

This case is interesting. In the absence of cystography, spontaneous bladder rupture is less convincing. Why a cystography (conventional CT cystography or preferably an ultra-low-dose CT cystography) was not performed?

Also, this manuscript has too many spell and grammatical errors and also abbreviations that need to be spelled out. 

Author Response

Response to Reviewer 1 Comments

Please see the attachment file (revised manuscript). 

Point 1:  
This case is interesting. In the absence of cystography, spontaneous bladder rupture is less convincing. Why a cystography (conventional CT cystography or preferably an ultra-low-dose CT cystography) was not performed?

Response: Thank you for your comments.

AS you mentioned, cystography is useful for the diagnosis of spontaneous bladder rupture. We will try to perform it aggressively for future cases.

In this case, a cystoscopy was performed just after the inflammatory findings improved. However, the patient complained of severe abdominal pain during cystoscopy. Then, we stopped the cystoscopy. Considering the invasiveness, we could not perform additional cystography. Therefore, our diagnosis was based on a comprehensive evaluation of cystoscopy findings, ascites findings, and CT findings.

We added this information and indicated the corrections with a blue highlight in the main text of “Case Report” (page 2, line 95), and “Discussion” (page 5, line 150 – 154). Please see the attachment file "revised manuscript".

Point 2:
Also, this manuscript has too many spell and grammatical errors and also abbreviations that need to be spelled out. 

Response: Thank you for your comment. This manuscript has been already edited by English language editing service “Editage”. We will also submit a “Certificate of editing” file to the editorial. We added some abbreviations that need to be spelled out. We indicated the corrections with a blue highlight in the main text of “Abbreviations” (page 1, line 22 – 24).

Reviewer 2 Report

  1. Abstract should include the management done to this patient i.e. long term foleys catheter.
  2. The phrase "panperitonitis' is not appropriate common universal phrase. Most people would accept as "generalized peritonitis" rather than pan peritonitis. So i suggest to edit this. This is also a case of secondary peritonitis and this terminology of primary, secondary, tertiary peritonitis should be added somewhere in discussion section.
  3. I dont see any biological mechanisms included. What does radiotherapy do - how does it lead to damage to urinary bladder - what is the mechanism - pls include this, the cellular changes/mechanisms. 
  4. table 1 and 2 = has to be deleted. This is not how you present investigations data. Include relevant, pertinent, in the text of case description. 
  5. Omit the Figure 1 as it just shows ascites. Figure 2 also shows that same and thus, we will trust that there was no free air in figure 1 too. Delete that figure.
  6. Cystoscopy - is there a better image? 
  7. You state that patient declined surgery. What surgery was proposed - what was the thought, plan? I know you didnt operate - but what was the thought process? This needs to be included.
  8. How often you change the foleys catheter? what are the risks of long term catheter? Whats her predicted survival from gyne cancer aspect? Pls give such data
  9. The case mentions peritonitis. I dont see any discussion on management of sepsis, general principles of treating sepsis - PMID: 34563232 , PMID: 31341511 etc

Author Response

Response to Reviewer 2 Comments

Please see the attachment file (revised manuscript).

Point 1:  
Abstract should include the management done to this patient i.e. long term foleys catheter.

Response: Thank you for your comments.

We added a sentence about the management for spontaneous bladder rupture (SBR) in the “Abstract” section and indicated the corrections with a yellow highlight in the main text of “Abstract” (page 1, line 33).

Point 2:
The phrase "panperitonitis' is not appropriate common universal phrase. Most people would accept as "generalized peritonitis" rather than pan peritonitis. So I suggest to edit this. This is also a case of secondary peritonitis and this terminology of primary, secondary, tertiary peritonitis should be added somewhere in discussion section.

Response: Thank you for your comment.

As you mentioned, we changed the term “panperitonitis” to “generalized peritonitis” throughout the manuscript.

In addition, we discussed about primary, secondary, and tertiary peritonitis. We added this information and indicated the corrections with a yellow highlight in the main text of “Discussion” (page 4, line 118 – 120), in the main text of “References” (page 7, line 240), and with a green highlight (including response to reviewer 3) in the main text of “Introduction” (page 2, line 50 – 55).

Point 3:
I don’t see any biological mechanisms included. What does radiotherapy do - how does it lead to damage to urinary bladder - what is the mechanism - pls include this, the cellular changes/mechanisms. 

Response: Thank you for your comments.

We described some biological mechanisms (anatomical and pathological perspectives) for SBR along with previous reports.

We added this information and indicated the corrections with a yellow highlight in the main text of “Discussion” (page 4, line 125 – 132), “References” (page 7, line 245 – 247).

Point 4:  
Table 1 and 2 has to be deleted. This is not how you present investigations data. Include relevant, pertinent, in the text of case description. 

Response: Thank you for your comments.

As you mentioned, we deleted Table 1 and 2. Instead of this, we described only relevant laboratory findings. We added this information and indicated the corrections with a yellow highlight in the main text of “Case Report” (page 2, line 64 – 66, 72 - 73, and 76 - 80).

Point 5:  
Omit the Figure 1 as it just shows ascites. Figure 2 also shows that same and thus, we will trust that there was no free air in figure 1 too. Delete that figure.

Response: Thank you for your comments.

As you mentioned, we deleted Figure 1. We changed the number of Figure 2-4.

Point 6:  
Cystoscopy - is there a better image?

Response: Thank you for your comments.

There is nothing better than the image submitted. Since the patient complained of severe abdominal pain during cystoscopy, a temporary perforation due to intravesical pressure increasing was suspected. We stopped the cystoscopy. Therefore, our diagnosis was based on a comprehensive evaluation of cystoscopy findings, ascites findings, and CT findings. We described about this in the main text of “Discussion”. We indicated the corrections with a blue highlight (including response to reviewer 1) in the main text of “Discussion” (page 5, line 150 – 154).

Point 7:  
You state that patient declined surgery. What surgery was proposed - what was the thought, plan? I know you didn’t operate - but what was the thought process? This needs to be included.

Response: Thank you for your comments.

The surgeon suggested suturing and omentoplasty to the wall thinning in the dome of the bladder. However, the patient had a history of suffering, during abdominal surgery for uterine endometrial cancer. Although we explained the risks of conservative treatment with long-term bladder catheter placement (the risk of urinary tract infection and recurrent bladder rupture), the patient declined the surgery. We added this information and indicated the corrections with a yellow highlight in the main text of “Case Report” (page 2, line 98 – page 3, line 100).

Point 8:  
How often you change the foleys catheter? What are the risks of long-term catheter? What is her predicted survival from gyne cancer aspect? Pls give such data.

Response: Thank you for your comments.

The frequency of exchange the bladder catheter was once a month. Since the prognosis of uterine endometrial cancer has been no recurrence for 21 years, we considered that it would not affect the survival of this patient. Considering the overall survival of this patient, surgery might be preferable for the patient, however, the patient declined the surgery because of the invasiveness, as mentioned in the response to Point 7. We added a reference for the risk of long-term bladder catheter placement.

We added this information and indicated the corrections with a yellow highlight in the main text of “Discussion” (page 5, line 160 – 164), and “References” (page 8, line 273).

Point 9:  
The case mentions peritonitis. I don’t see any discussion on management of sepsis, general principles of treating sepsis - PMID: 34563232, PMID: 31341511 etc.

Response: Thank you for your comments.

We discussed about the sepsis caused by generalized peritonitis. We added references as you recommend. We indicated the corrections with a yellow highlight in the main text of “Discussion” (page 5, line 165 – 170) and “References” (page 8, line 274 – 277).

Reviewer 3 Report

Dear Authors,

The manuscript submitted by Watanabe T. et al., entitled "A rare case of recurrent panperitonitis caused by spontaneous urinary bladder rupture after radiotherapy: A case report and literature review," is interesting.

The English need significant improvements before it can be processed further. Here are my comments:

Please read and apply the requirements of the journal - https://www.mdpi.com/journal/medicines/instructions .

  1. Please arrange affiliations of the authors according to the journal requirements.
  2. Please delete from the abstract background, methods, and result, discussion and conclusion.
  3. l. 28 should be diagnosed with panperitonitis.
  4. The Introduction section needs to be revised; it is too short. The authors should try to include the causes of SBR and the rarity of panperitonitis produced by SBR. There is no information about the diagnosis and treatment of peritonitis.
  5. Please include in case report information regarding uroculture.
  6. l. 67, please detail the conservative treatment, except for antibiotics
  7. l. 71 it should be another paragraph
  8. l. 72, please describe the 3rd episode
  9. Table 1 should be revised and include the data from 1st day of all admissions. The tables should be realized in Office Word, not as a picture.
  10. l. 122-124 The authors have not present any information about the uroculture of the patient or the diagnostics of cystitis.
  11. l.142-143, the authors should include the value of Cre and BUN for all admissions.
  12. l.143-146 needs a reference

Author Response

Response to Reviewer 3 Comments

Please see the attachment file (revised manuscript).

Point 1:  
Please arrange affiliations of the authors according to the journal requirements.

Response: Thank you for your comments.

We changed our affiliations according to the journal guideline. We indicated the corrections with a green highlight in the main text of “Title page” (page 1, line 7 - 12).

Point 2:
Please delete from the abstract background, methods, and result, discussion and conclusion.

Response: Thank you for your comment.

As you mentioned, we deleted them.

Point 3:
Line 28 should be diagnosed with panperitonitis.

Response: Thank you for your comments.

In accordance with the comment on Reviewer 2, we corrected the term “generalized peritonitis”.

We indicated the corrections with a yellow highlight (including response to reviewer 2) in the main text of “Abstract” (page 1, line 29 – 31).

Point 4:  
The Introduction section needs to be revised; it is too short. The authors should try to include the causes of SBR and the rarity of panperitonitis produced by SBR. There is no information about the diagnosis and treatment of peritonitis.

Response: Thank you for your comments.

We added some information about SBR in “Introduction” section. We indicated the corrections with a green highlight in the main text of “Introduction” (page 2, line 50 – 55).

Point 5:  
Please include in case report information regarding uroculture.

Response: Thank you for your comments.

As you mentioned, we added the information regarding uroculture and urinalysis. In accordance with Reviewer 2 comments, we deleted Table 1 (including urinalysis). Instead of this, we added the urinalysis findings along with uroculture.

We indicated the corrections with a green highlight in the main text of “Case Report” (page 2, line 93 - 94).

Point 6:  
Line 67, please detail the conservative treatment, except for antibiotics.

Response: Thank you for your comments.

We described the details about the conservative treatment. Since the general condition of the patient improved rapidly, we performed only intravenous fluid treatment in addition to antibiotics. We indicated the corrections with a green highlight in the main text of “Case Report” (page 2, line 82 – 83).

Point 7:  
Line 71, it should be another paragraph.

Response: Thank you for your comments.

We changed as you mentioned (page 2, line 89).

Point 8:  
Line 72, please describe the 3rd episode.

Response: Thank you for your comments.

We added the 3rd episode of generalized peritonitis. We indicated the corrections with a green highlight in the main text of “Case Report” (page 2, line 87 – 89).

Point 9:  
Table 1 should be revised and include the data from 1st day of all admissions. The tables should be realized in Office Word, not as a picture.

Response: Thank you for your comments.

We deleted Table 1 in accordance with Reviewer 2 comments. Instead of this, we described only relevant laboratory findings in the main text of “Case Report”. We added this information and indicated the corrections with a yellow highlight (including response to reviewer 2) in the main text of “Case Report” (page 2, line 64 – 66, 72 – 73, and 76 - 80).

Point 10:  
Line 122-124, the authors have not present any information about the uroculture of the patient or the diagnostics of cystitis.

Response: Thank you for your comments.

We added the information about the uroculture and urinalysis of the patient. We indicated the corrections with a green highlight in the main text of “Case Report” (page 2, line 93 – 94), and “Discussion” (page 4, line 135 - 140).

Point 11: 
Line 142-143, the authors should include the value of Cre and BUN for all admissions.

Response: Thank you for your comments.

We added the serum level of BUN and Cre for all admissions. We indicated the corrections with a green highlight in the main text of “Discussion” (page 5, line 172 - 175).

Point 12: 
Line 143-146 needs a reference. 

Response: Thank you for your comments.

We added reference as you recommended. We indicated the corrections with a green highlight in the main text of “References” (page 8, line 278 - 280).

Round 2

Reviewer 1 Report

The revised manuscript is much better. However, the manuscript still has many grammatical errors and unconventional language style throughout the manuscript which is harder to understand. I would recommend a thorough language editing before acceptance. 

I have included the following sentences as examples. 

Line 133 - "SBR can occur secondary peritonitis ......"  This sentence is difficult to follow. I would recommend to start the discussion "SBR is a rare cause of secondary peritonitis......."

Line 184 - "Although the sepsis-related organ failure assessment score of this patient is three points due to the renal failure....." - Please clarify which scoring system is referred here. 

Author Response

Response to Reviewer 1 Comments (Round 2)

Point 1:

The revised manuscript is much better. However, the manuscript still has many grammatical errors and unconventional language style throughout the manuscript which is harder to understand. I would recommend a thorough language editing before acceptance. 

I have included the following sentences as examples. 

Line 133 - "SBR can occur secondary peritonitis ......"  This sentence is difficult to follow. I would recommend to start the discussion "SBR is a rare cause of secondary peritonitis......."

Response: Thank you for your comments. Since the editorial office required us to resubmit this revised manuscript within 2 days in this time, it is difficult re-edit by English language editing service. We asked the editorial office to give us enough time to resubmit this manuscript after English language editing service. If so, we will resubmit the re-editted manuscript after that.

Furthermore, as you recommended, we changed the sentence. We indicated the corrections with a blue highlight in the main text of “Discussion” (page 4, line 133 – 135).

Point 2:  
Line 184 - "Although the sepsis-related organ failure assessment score of this patient is three points due to the renal failure....." - Please clarify which scoring system is referred here. 

Response: Thank you for your comments.

We calculated the score in accordance with Reference 40. We indicated the corrections with a blue highlight in the main text of “Discussion” (page 5, line 185) and “References” (page 9, line 322 - 323).

Reviewer 3 Report

Dear Authors,

Thank you to the authors for submitting a thoroughly revised version of their manuscript. The manuscript is vastly improved - particularly the Case Report and Discussion sections.

I feel that many language edits are still needed (l. 29 diagnosed with generalized peritonitis; ). Please check carefully for the use of scientific language and correct phrases. I suggest a native English speaker should review the manuscript.

The Introduction section is still too short. There is no information about the diagnosis and treatment of peritonitis. L. 51 one of the causes of SBR is trauma. L. 54-55 should be reformulated. 

L. 84 should be another paragraph.

Please describe the uroculture, if it was performed, for the first three admissions. 

l. 97-102 should be reformulated 

l. 118 SBR can occur secondary peritonitis???

l. 137 uroculture showed Klebsiella oxytoca and WBC??? (on uroculture, there is no WBC)

l. 165-170 should be deleted or reformulated. Why the authors suspected the sepsis, based on what???

l.195 SBR in the differential diagnosis or just causes of generalized peritonitis?

l. 209, please delete differential

I suggest the authors read the whole manuscript and correct it carefully. There are a lot of phrases with no sense and are hard to understand.

The Authors Contribution should be made according to journal requirements.

Author Response

Response to Reviewer 3 Comments (Round 2)

Point 1:

Thank you to the authors for submitting a thoroughly revised version of their manuscript. The manuscript is vastly improved - particularly the Case Report and Discussion sections.

Response: Thank you for your comments.

Point 2:

I feel that many language edits are still needed (l. 29 diagnosed with generalized peritonitis). Please check carefully for the use of scientific language and correct phrases. I suggest a native English speaker should review the manuscript.

Response: Thank you for your comments.

As you mentioned, we corrected the sentence (Line 29). We indicated the corrections with a red highlight in the main text of “Abstract” (page 1, line 29). This manuscript has been already edited by English language editing service “Editage”. We also submitted a “Certificate of editing” file to the editorial office. Since the editorial office required us to resubmit this revised manuscript within 3 days, it is difficult re-edit by English language editing service “Editagage”. If the editorial office gives us enough time to resubmit this manuscript, we will consider re-editing.

Point 3:  
The Introduction section is still too short. There is no information about the diagnosis and treatment of peritonitis. Line 51 one of the causes of SBR is trauma. Line 54-55 should be reformulated. 

Response: Thank you for your comments.

We added some information about diagnosis and treatment of peritonitis in “Introduction” section along with further references. We added a term “trauma” in the sentence as a cause of SBR (page 2, line 51) We indicated the corrections with a red highlight in the main text of “Introduction” (page 1, line 43 – page 2, line 70) and “References” (page 8, line 261 - 274). 

Point 4:

Line 84 should be another paragraph.

Response: Thank you for your comments.

We changed as you mentioned (page 3, line 100).

Point 5:

Please describe the uroculture, if it was performed, for the first three admissions.

Response: Thank you for your comments.

Since there were no urinary symptoms such as dysuria during the first three admissions, we did not perform uroculture. We learned that if we had performed uroculture from the beginning as you suggested, it might have helped in the diagnosis of spontaneous bladder rupture. 

Point 6:

Line 97-102 should be reformulated. 

Response: Thank you for your comments.

As you mentioned, we reformulated the manuscript. We indicated the corrections with a red highlight in the main text of “Case Report” (page 3, line 115 – 117).

Point 7:

Line 118 SBR can occur secondary peritonitis???

Response: Thank you for your comments. 

Reviewer 2 pointed out that this case is a secondary peritonitis caused by SBR.

In accordance with Reviewer 2 comment, we described generalized peritonitis caused by SBR as a secondary peritonitis, because we thought it was peritonitis following a rupture of the bladder, like a perforation of the gastrointestinal perforation.

Point 8:

Line 137 uroculture showed Klebsiella oxytoca and WBC??? (on uroculture, there is no WBC)

Response: Thank you for your comments.

As you mentioned, we deleted “and WBC”. WBC was only detected in the urinalysis (page 3, line 108 - 109).

Point 9:

Line. 165-170 should be deleted or reformulated. Why the authors suspected the sepsis, based on what???

Response: Thank you for your comments.

As you mentioned, we corrected some sentences. We indicated the corrections with a red highlight in the main text of “Case Report” (page 2, line 80) and “Discussion” (page 5, line 182 - 183).

Point 10:

Line 195 SBR in the differential diagnosis or just causes of generalized peritonitis?

Response: Thank you for your comments.

We consider that the most important thing is to know the disease and clinical features of SBR as a differential diagnosis. We indicated the corrections with a red highlight in the main text of “Discussion” (page 6, line 211 – 212). 

Point 11:

Line 209, please delete differential.

Response: Thank you for your comments.

As you mentioned, we deleted “differential”.

Point 12:
I suggest the authors read the whole manuscript and correct it carefully. There are a lot of phrases with no sense and are hard to understand.

The Authors Contribution should be made according to journal requirements.

Response: Thank you for your comment.

As we described above, if the editorial office gives us enough time to resubmit this manuscript, we will consider re-editing by English language editing service “Editage”.

Round 3

Reviewer 3 Report

Dear Authors,

The manuscript is improved. Congratulations on your work.

Author Response

The manuscript is improved. Congratulations on your work.

Responce: Thank you for your comment.